# Error-correcting dynamics in visual working memory

Matthew F. Panichello [1], Brian DePasquale[1], Jonathan W. Pillow[1,2] & Timothy J. Buschman [1,2]

Working memory is critical to cognition, decoupling behavior from the immediate world. Yet, it is imperfect; internal noise introduces errors into memory representations. Such errors have been shown to accumulate over time and increase with the number of items simultaneously held in working memory. Here, we show that discrete attractor dynamics mitigate the impact of noise on working memory. These dynamics pull memories towards a few stable representations in mnemonic space, inducing a bias in memory representations but reducing the effect of random diffusion. Model-based and model-free analyses of human and monkey behavior show that discrete attractor dynamics account for the distribution, bias, and precision of working memory reports. Furthermore, attractor dynamics are adaptive. They increase in strength as noise increases with memory load and experiments in humans show these dynamics adapt to the statistics of the environment, such that memories drift towards contextually-predicted values. Together, our results suggest attractor dynamics mitigate errors in working memory by counteracting noise and integrating contextual information into memories.

---

[1] Princeton Neuroscience Institute, Princeton University, Princeton, NJ 08540, USA. [2] Department of Psychology, Princeton University, Princeton, NJ 08540, USA. Correspondence and requests for materials should be addressed to T.J.B. (email: tbuschma@princeton.edu)

Working memory is our ability to maintain information without direct sensory input. It allows us to decouple behavior from the immediate world, serving as the substrate for planning and problem solving[1]. Despite its fundamental role in cognition, information in working memory is not stored with perfect fidelity. Errors accrue over time[2–5] and with the number of items simultaneously held in working memory[6–11].

Errors in working memory are thought to be due, in part, to noise in the neural representations underlying memories. Random noise can cause memory representations to diffuse away from their original state over time, leading to behavioral errors[12,13]. This is consistent with theoretical work that suggest memory representations are maintained in a continuum of stable states (known as a 'line' or 'ring' attractor[14–16]). Such systems can encode continuous variables with high precision and in an unbiased manner. This is important for many domains, such as visual working memory for color or orientation. However, a disadvantage of such systems is that they integrate noise: perturbations of representations along the stable continuum are maintained, resulting in a steady accrual of error over time. Because of this, variability in spiking activity places a bound on the accuracy of working memory representations[15].

In contrast, theoretical work has suggested the impact of noise can be mitigated if memories are stored using a finite set of stable states known as discrete attractors[17–21]. In such systems, memory representations drift towards the attractor states. Once there, memories are stable and therefore resistant to diffusive noise. However, this comes at the the cost of discretizing continuous information, reducing precision and inducing bias into memory.

Here we test whether the brain uses discrete attractor dynamics to mitigate the impact of noise on working memory. By fitting a flexible dynamical systems model to data from individual subjects, we estimate the forces governing the temporal evolution of working memory representations in both humans and monkeys. We show that discrete attractor dynamics better explain behavior than competing models of memory dynamics. Indeed, discrete attractor dynamics account for the distribution, bias, and precision of working memory reports and the accumulation of error in memory over time. Furthermore, these dynamics adapt to changes in context and memory load in a way that minimize errors in working memory.

## Results

**Systematic error in memory increases with load and time.** To understand the dynamics governing working memory representations, we examined the behavior of humans ($N = 90$) and monkeys ($N = 2$) performing a delayed estimation task[22] (Fig. 1a). Subjects were instructed to remember the color of 1 to 3 simultaneously-presented stimuli located at different positions on the display (humans saw 1 or 3 items; monkeys saw 1 or 2). After a variable memory delay, subjects reported the remembered color at a cued target location using a continuous scale. Stimulus colors were drawn uniformly from an isoluminant circular color space. We quantified error as the angular deviation between the target color and the subject's report. As expected[2–11], the average absolute error increased as a function of delay and working memory load in both humans and monkeys (Fig. 1b; humans (H): load, $F(1, 89) = 147.23$, $p < 1 \times 10^{-15}$; delay, $F(1, 89) = 85.44$, $p = 1.17 \times 10^{-14}$; load x delay, $F(1, 89) = 13.92$, $p = 3.36 \times 10^{-4}$, analysis of variance; monkey W (W): load, $p < 0.001$; delay, $p = 0.006$; load x delay, $p = 0.495$, bootstrap; monkey E (E): load, $p < 0.001$; delay, $p = 0.009$; load x delay, $p = 0.303$, bootstrap).

Despite the uniform distribution of target colors, the responses of both human and monkey subjects were significantly non-

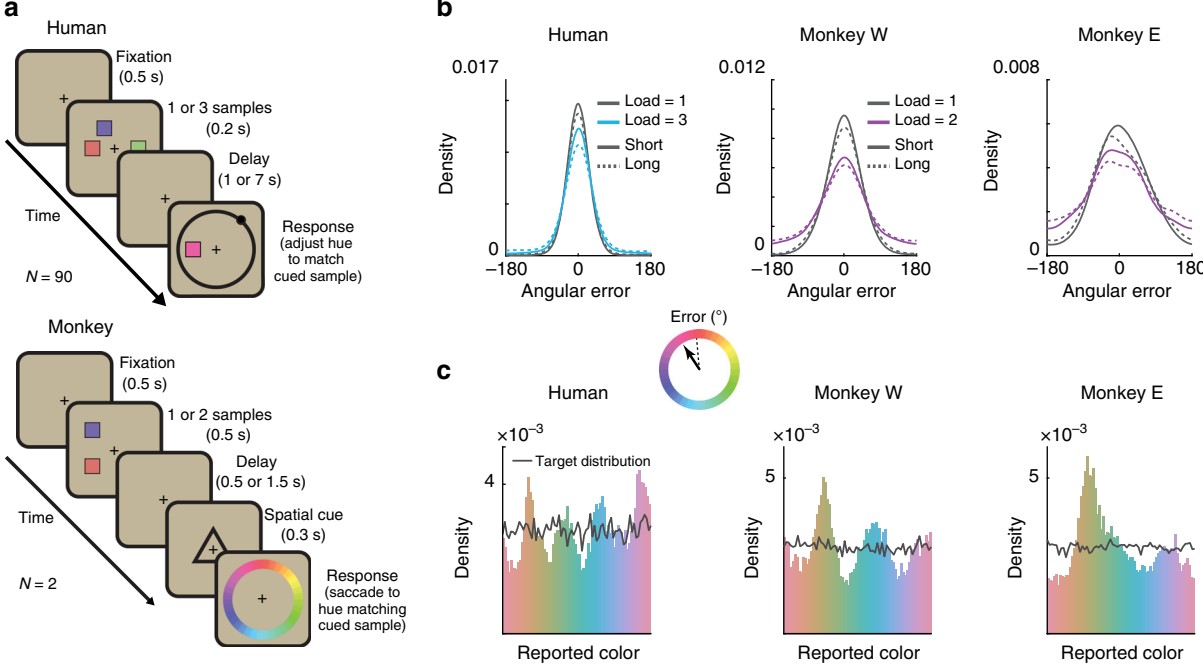

**Fig. 1** Memories cluster in a continuous working memory task. **a** Top: humans ($N = 90$) performed a color delayed-estimation task in which they reported the color of a spatially-cued sample after a variable delay. Humans made their report by adjusting the hue of the response probe by rotating a response wheel (black circle) using a mouse. We rotated the mapping between wheel angle and color on each trial to avoid spatial encoding of color memories. Bottom: monkeys ($N = 2$) performed a similar task. A symbolic cue indicated which sample to report (top or bottom). Monkeys reported a specific color value using an eye movement to a color wheel that was rotated on each trial. **b** Distribution of angular error for humans (top) and monkeys (bottom). Error increased with load and delay time. Gray lines = low load, blue lines = high load, solid lines = short delay, dashed lines = long delay. Inset: Error is calculated as the angular deviation between the color of the cued sample and the reported color in color space. **c** Non-uniform distribution of reported colors for humans (top) and monkeys (bottom). Gray line shows the distribution of target colors. Source data are provided as a Source Data file

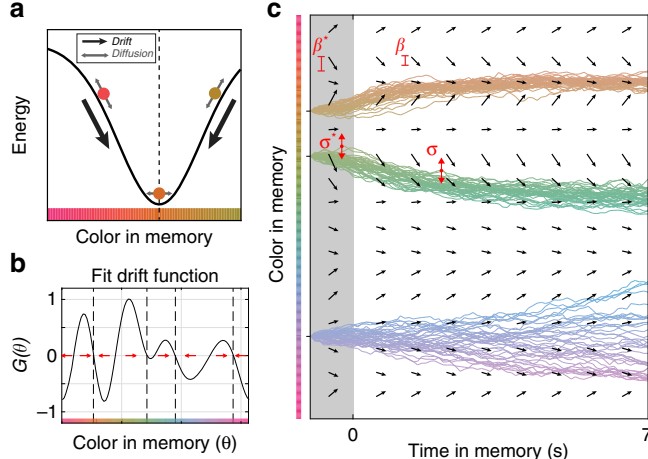

**Fig. 2** Structure of dynamical model. **a** Illustration of the influence of attractors on color memory. Attractors (dashed line) cause memories to drift over time (black arrow), introducing bias in reports. Noise causes memories to randomly diffuse (grey arrows). **b** The drift function $G(\theta)$ describes how a memory will change based on its current state. Red arrows show the direction of drift; attractors have converging drift (dashed lines). We estimated $G(\theta)$ for each subject using a linear combination of von mises derivatives. **c** The simulated evolution of three color memories during a hypothetical trial. Memory evolves over time according to the drift function (vector field) and random noise. Each line indicates the temporal evolution of a remembered color under a different realization of the noise process. Terms described in main text. Source data are provided as a Source Data file

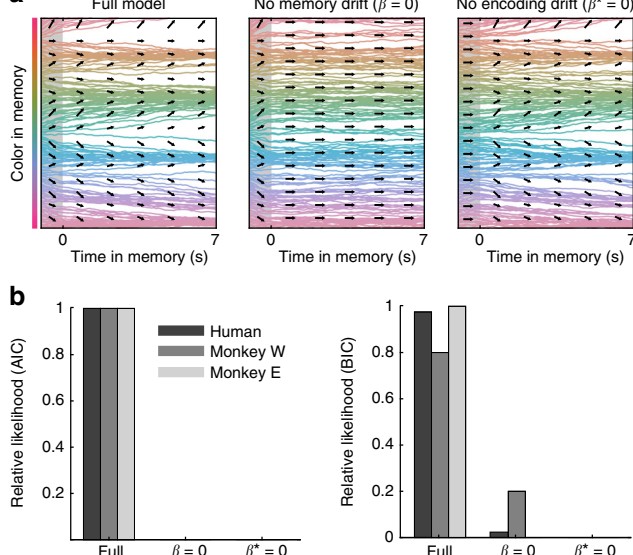

**Fig. 3** Behavior is best explained by attractor dynamics during encoding and memory. **a** Simulated memory trajectories from the best-fitting model for one subject. Left: The full dynamical model includes drift towards attractor states during both encoding and memory. Reduced models include drift only during encoding (middle) or memory (right). **b** AIC and BIC model weights (normalized relative likelihood) for the full model compared with models with zero drift during encoding ($\beta^* = 0$) and memory ($\beta = 0$). Values indicate the probability that the given model is the best model in the set[49]. Source data are provided as a Source Data file

uniform[23–26] (Fig. 1c, humans and monkeys $p < 0.001$ against uniformity, Hodges-Ajne test; $p < 0.001$ against target distribution, permuted Kuipers test). This was reflected in a significant decrease in the entropy of the response distribution relative to the target distribution (H: 2.54 vs. 2.61 bits, $t(89) = 13.90$, $p < 1 \times 10^{-15}$, $t$ test; W: 2.61 vs. 2.65 bits, $p < 0.001$, bootstrap; E: 2.58 vs. 2.65 bits, $p < 0.001$, bootstrap). Responses clustered around specific colors, seen as peaks in the response histogram (Fig. 1c). Clustering increased with delay time ($F(1, 89) = 9.56$, $p = 0.003$, analysis of variance) and with memory load in humans ($F(1, 89) = 5.45$, $p = 0.022$; Supplementary Figs 1 and 2), suggesting that clustering is the result of a load-dependent dynamic process that unfolds over the course of encoding and the memory delay.

**Attractor dynamics influence memory representations.** Motivated by these results, we tested the hypothesis that discrete attractor dynamics underlie the evolution of working memory representations. Attractor states can be conceptualized as local minima in an energy landscape over mnemonic (color) space, such that memories drift towards nearby attractors over time (Fig. 2a). These dynamics could provide a mechanistic explanation for the observed clustering of memory reports.

To test for the existence of discrete attractors, we developed a model to characterize the dynamics governing working memory representations. The model describes memory error as a combination of diffusion from noise in the neural representation[12,14,15] and drift towards attractor states. Diffusion was quantified as a random walk from the current location in mnemonic space with no bias ($\mu = 0$) and a variance ($\sigma_L^2$) that depended on the number of colors presented ($L$ = memory load). Discrete attractor dynamics were modeled by fitting a function $G(\theta)$ that describes how a remembered color $\theta$ will drift as a function of its current value (Fig. 2b). Positive drift values reflect a

clockwise drift (to the right in Fig. 2b) while negative values reflect a counterclockwise drift (to the left). Thus, attractors are points in mnemonic space that 1) are fixed, such that they have no drift, and 2) pull nearby memories towards themselves, indicated by a negative slope in the drift function (Fig. 2b, dashed lines). Subjects displayed the same number and location of clusters in their distribution of memory reports regardless of load condition (Supplementary Fig. 2), so we assumed that the pattern of drift did not vary with load (i.e., the shape of the function $G(\theta)$ was the same across loads). However, as with diffusion, the strength of the drift was allowed to vary across memory load (i.e. $G(\theta)$ is scaled by $\beta_L$).

Together, drift and diffusion define the temporal evolution of memories during the delay (Fig. 2c, $d\theta = \beta_L G(\theta)dt + \sigma_L \mathcal{N}(0, dt)$). Previous work has shown that reports of perceived colors are clustered, although clustering is greater for colors held in working memory[24]. To capture clustering and other sources of error[27,28] that emerge during encoding, inputs were first passed through an encoding stage governed by a similar drift and diffusion process with the same drift function $G(\theta)$. However, the strength of drift and diffusion during encoding was set independently by two additional parameters ($\beta_L^*$ and $\sigma_L^*$; see "Methods" section for details). This allowed us to test for discrete attractor dynamics during both encoding and the memory delay (Fig. 3). Finally, three additional terms in the model captured errors due to forgetting of memories[7], responses to non-targets[8], and noise introduced at decoding (see Methods for details). Model parameters were estimated by maximizing the joint likelihood of the observed memory reports across individual trials for each subject. Critically, the model did not assume attractor dynamics (Fig. 3a); when $\beta_L$ and $\beta_L^*$ are zero, memories are only influenced by diffusion, forgetting, and responses to non-targets, as in previous models.

Discrete attractor dynamics provide a better account of behavior than models in which memories only diffuse randomly

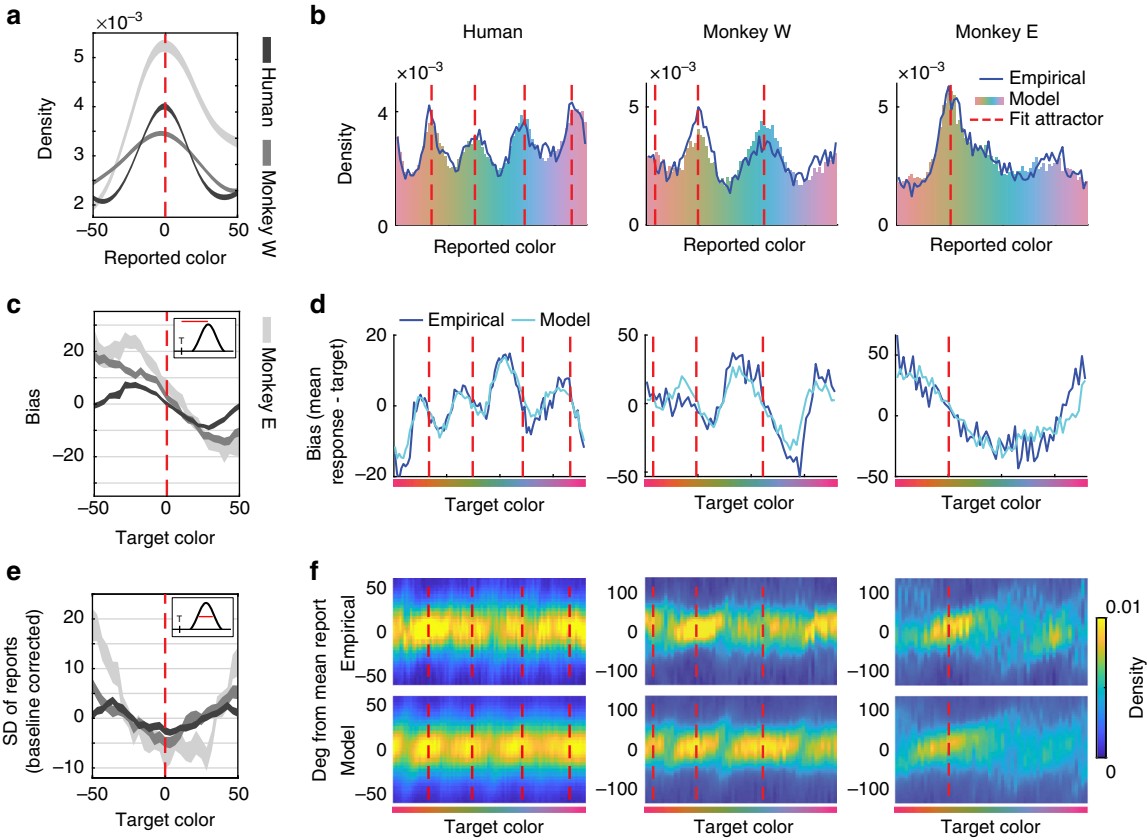

**Fig. 4** Attractors explain the distribution, bias, and precision of working memory reports. **a** Observed distribution of memory reports around fit attractors (red dashed line). *X*-axis: degrees in color space. **b** Distribution of simulated memory reports from the full model fit to human and monkey subjects. Blue line shows empirical distribution of reports. **c** Observed bias +/− SEM around fit attractors. Inset: bias is calculated as the angular distance between the target and mean report. Positive values indicate clockwise (CW) drift; negative values indicate counter-clockwise (CCW) drift. **d** Mean bias in reports as a function of target color (dark blue line), with predictions from best-fit models (light blue line). **e** Observed SD of reports +/− SEM around fit attractors. **f** Distribution of memory reports around their mean as a function of target color (top row), with predictions from best-fit models (bottom row). More precise memory reports are indicated by tighter distributions around their mean. Source data are provided as a Source Data file

(Fig. 3). To demonstrate this, we compared the full model with drift and diffusion to reduced models without drift towards attractor states during encoding or memory ($\beta^{\star} = 0$ or $\beta = 0$, Fig. 3a). Three model comparison statistics (AIC, BIC, and cross-validated likelihood) all indicated that the full model performed best (Fig. 3b and Supplementary Tables 1 and 2; H: relative likelihood of full model = 1.00 (AIC) and 0.98 (BIC); W: 1.00 and 0.80; E: 1.00 and 1.00). Thus, both the encoding and delay periods are characterized by drift of memories towards attractor states.

As seen in previous work[23,24], memory reports clustered at certain points in color space, and the bias and precision of reports vary systematically around points of peak clustering. Here, we show that the discrete attractor dynamics explain these variations. First, discrete attractor dynamics predict a clustered distribution of memory reports because memories tend to accumulate at attractor states. Accordingly, colors near attractor states identified by each subject's best-fit model were reported more frequently than average (Fig. 4a, H: $t(89) = 43.49$, $p = 9.54 \times 10^{-62}$; W: $p < 0.001$, bootstrap; E: $p < 0.001$, bootstrap). The distribution of memory reports predicted by each subject's best-fit model provides an excellent fit of the empirically observed distribution of memory reports (Model: Fig. 4b, H: $r(70) = .909$, $p = 2.57 \times 10^{-28}$; W: $r(70) = .741$, $p = 9.93 \times 10^{-14}$; E: $r(70) = .934$, $p = 4.21 \times 10^{-33}$, Pearson's $r$).

Second, discrete attractors explain bias in working memory reports. Memories of a particular target color will consistently drift towards the closest attractor state, inducing systematic bias

in subjects' reports. This is evident in subjects' behavior: memories for target colors counter-clockwise to an attractor location tended to drift clockwise, while targets clockwise to an attractor tended to drift counter-clockwise (Fig. 4c, H: mean slope −0.40 less than zero, $t(89) = -12.60$, $p = 1.73 \times 10^{-21}$, $t$ test; W: −0.59, $p < 0.001$, bootstrap; E: −0.73, $p < 0.001$, bootstrap). Model-free analyses showed similar effects. The peaks in the response histogram provide independent estimates of attractor locations. Aligning the bias around peaks in the response histogram reveals a similar pattern with a negative slope (Supplementary Fig. 3[23,24]). Furthermore, the model provides a good qualitative fit to the pattern of bias across color space (Fig. 4d). The model's predicted pattern of biases for each target color was highly correlated with the empirically observed pattern of biases in both human and monkeys (H: $r(88) = 0.939$, $p = 1.41 \times 10^{-42}$; W: $r(58) = 0.864$, $p = 6.95 \times 10^{-19}$; E: $r(58) = .850$, $p = 8.13 \times 10^{-18}$, Pearson's $r$).

Third, discrete attractors explain the precision of working memory reports. Memories near attractors are more stable: as diffusive noise drives a memory representation away from an attractor, drift will pull it back towards the attractor, resulting in a narrow response distribution. For both humans and monkey subjects, the standard deviation (SD) of memory reports was lower for targets near attractors identified by each subject's best-fit model (Fig. 4e, H: $\Delta SD = -1.96$, $t(89) = -4.90$, $p = 4.20 \times 10^{-6}$, $t$ test; W: −2.96, $p < 0.001$, bootstrap; E: −5.59, $p < 0.001$, bootstrap). Model-free analyses again showed similar effects: SD

was significantly reduced at the peaks in the response histogram (Supplementary Fig. 3). As with bias, discrete attractor dynamics predict the pattern of precision across color space (Fig. 4f). The model's predicted pattern of precision as a function of target color was correlated with the empirically observed values in both human and monkeys (Fig. 4f, H: $r(88) = .370$, $p = 3.27 \times 10^{-4}$; W: $r(58) = .377$, $p = 0.003$; E: $r(58) = .630$, $p = 6.88 \times 10^{-8}$, Pearson's $r$).

We can exclude several other possible explanations for the non-uniform distribution of memory reports. One alternative explanation is that clustering is driven by subjects guessing with a biased distribution on a subset of trials. However, if true, then bias would not display an 'attractive' positive-to-negative transition at cluster peaks and precision would not depend on the identity of the item in memory (Supplementary Fig. 4). A second alternative is that clustering could be driven by a nonlinear mapping between the stimulus space chosen by the experimenter and the subject's true perceptual space. However, such a model predicts the opposite pattern of bias across color space (Supplementary Fig. 5; see "Methods" section for details).

The discrete attractor model also predicts how errors in working memory evolve over time. First, the discrete attractor model accurately recapitulates the increase in error over the delay. To measure the change in mean error over the delay, we measured error for memory delays ranging from 1 to 7 s (Experiment 1b; Supplementary Fig. 6a; 120 new human subjects). The discrete attractor model provided a good fit to the increase in error with memory delay (Supplementary Fig. 6b).

Second, the discrete attractor model makes the specific prediction that memories of different target colors are expected to accumulate error at different rates. Attractors are 'stable fixed points' because they counteract perturbations of memory due to random noise. Perturbations are corrected by drift back towards the stable fixed point. Because this process occurs continuously over time, memories of target colors near stable fixed points are not only more precise overall (i.e., as in Fig. 4e), but also accumulate error at a relatively slow rate over time (Fig. 5a). In contrast, target colors near 'unstable fixed points' accumulate error relatively quickly over time because random perturbations away from these points are exacerbated by drift away from the unstable fixed point (Fig. 5a). To test this prediction, we first identified stable and unstable fixed points for each subject by identifying target colors with attractive bias (zero with a negative slope) or repulsive bias (zero with a positive slope). We then calculated how much error increased on long delay trials relative to short delay trials for target colors near stable and unstable fixed points. For both humans ($p = 0.036$, bootstrap) and monkeys (W: $p < 0.0001$, E: $p = 0.024$, bootstrap), error increased more over time for target colors near putative unstable fixed points (Fig. 5b).

**Attractor dynamics strengthen with load**. The error-correcting properties of attractors may be especially critical when memory load is high. High memory load decreases the magnitude of neural responses[27], which is thought to render memories more susceptible to noise and, therefore, increase diffusion[15,29]. Indeed, as estimated by the model fits to experiment 1a, diffusion during the memory delay increased with memory load (Fig. 6a, $\sigma_L^2$, H: $p = 0.001$; W: $p = 0.021$, E: $p = 0.010$, bootstrap) although changes during encoding were mixed (Fig. 6b, $\sigma_L^{2*}$, H: $p < 0.001$; W: $p = 0.459$, E: $p = 0.100$, bootstrap). Consistent with the theory that attractor dynamics compensate for diffusion, we saw a commensurate increase in drift during the memory delay (Fig. 6c, $\beta_L$, H: $p = 0.002$; W: $p = 0.026$, E: $p = 0.026$, bootstrap) and during encoding (Fig. 6d, $\beta_L^*$, H: $p = 0.001$; W: $p = 0.024$, E: $p = 0.009$, bootstrap). Similarly, two model-free measures of drift, clustering

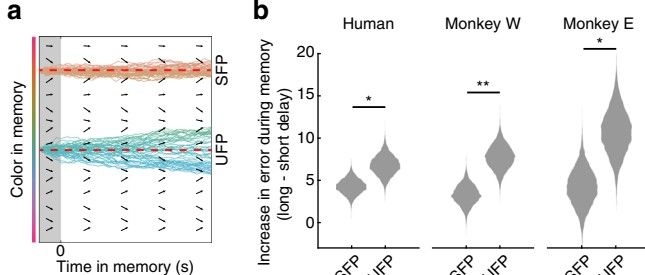

**Fig. 5** Memories near attractors are more stable. **a** If discrete attractors underlie working memory, then memories of different target colors will accumulate error at different rates over time. Memories of target colors near stable fixed points (SFPs) accumulate a relatively small amount of error over time because perturbations away from SFPs due to random noise are corrected by drift back towards the SFP. Unstable fixed points (UFPs) lie in between attractors; perturbations away from UFPs due to random noise are exacerbated by drift away from the UFP. Red dashed lines the indicate location of two of the fixed points. **b** Mean increase in error at SFPs and UFPs identified by the fit model. Distributions reflect bootstrapped values. *$p < 0.05$, **$p < 0.01$, bootstrap. Source data are provided as a Source Data file

of responses and mean absolute bias, increased with load (Supplementary Fig. 1). Note that although the rate of drift and diffusion during memory is less than that during encoding, their effects accumulate over the course of the memory delay.

**Experience modifies the location of attractor states**. While discrete attractors compensate for diffusion, they also induce systematic error into working memory. Thus, there is a trade-off between the finite error caused by drifting toward an attractor and the ever-increasing error associated with diffusion. To test whether discrete attractors improved overall performance, we simulated memory dynamics for the full discrete attractor model ('drift + diffusion') and from the same model with encoding and memory drift set to zero ('diffusion', $\beta = \beta^* = 0$). Thus, we can ask how memory accuracy would change if diffusion were held constant and we manipulated only the presence or absence of discrete attractor states. As shown in Fig. 7a, the two models accumulate error at different rates over time. Initially, the mean absolute error is greater in the drift + diffusion model due to memory corruption by drift towards attractor states during encoding and the early delay period ($p < 0.05$ for $t < 11$ s, bootstrap). However, discrete attractors also counteract diffusive noise and so, as the delay increases, the drift + diffusion model performs significantly better than the diffusion model ($p < 0.05$ for $t >= 33$ s, bootstrap), with the crossover in performance occurring at $t \sim 17$ s. Thus, attractor dynamics have a greater impact the longer information is held in working memory.

Discrete attractor dynamics are most beneficial when they adapt to the current context. For example, the statistics of many visual features in the real world are not uniform across perceptual space (including color[30]). In this case, errors can be reduced if attractor states reflect the statistics of the environment, such that attractors occur at the location of common stimuli. To demonstrate this, we tested the performance of the full discrete attractor model in different environments. Environments varied in the proportion of target colors drawn from within 10 degrees of an attractor. For example, when 50% of targets were drawn from nearby an attractor, the 'drift + diffusion' model significantly reduced working memory error for all $t$ (Fig. 7a, red trace). Parametrically varying the proportion of biased colors revealed that discrete attractor states tuned to the statistics of the

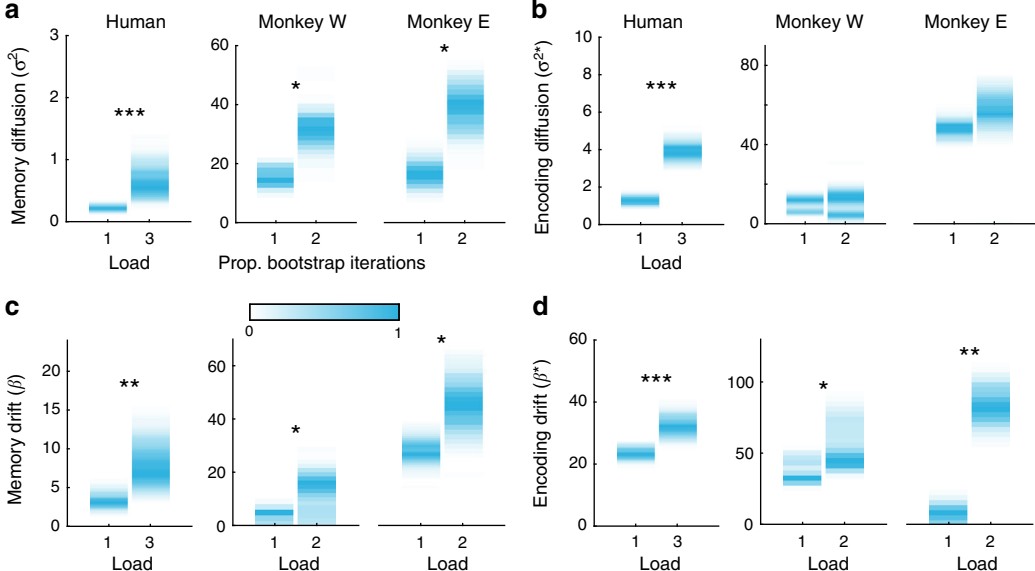

**Fig. 6** Drift and diffusion increase with memory load. Experiment 1a maximum likelihood parameter fits for the diffusion (**a**, **b**) and drift (**c**, **d**) scaling parameters during memory and encoding. Color intensity reflects normalized proportion of bootstrap iterations. As detailed in the Methods, all parameters are rates (change per second); dynamics during encoding evolve over a fixed period of time (simulated as 1 s), while memory dynamics evolve over the memory delay, which varied from trial to trial. *$p < 0.05$, **$p < 0.01$, ***$p < 0.001$, bootstrap. Source data are provided as a Source Data file

environment reduced memory error, even with modest biases in the color distribution (Fig. 7b). These results suggest that, in order to minimize working memory errors, attractor dynamics should adapt to the statistics of the current environment.

To test whether memory dynamics adapt to context, we collected data from 120 additional human subjects in a continuous working memory task with a biased stimulus distribution (Experiment 2, Fig. 7c). During this task, the statistics of the environment were such that half of all stimuli were drawn from one of four common colors (randomly chosen for each subject) while the other half were drawn from a uniform distribution.

Both model-free and model-based analyses suggest that participants developed attractor states at the common color locations. First, attractor states, as identified by fitting the dynamical model, were significantly more likely to occur at the location of common colors than expected by chance (Fig. 7d, $p < 0.001$, randomization test, model fits were limited to trials in which the target color was drawn from a uniform distribution). Second, consistent with the accumulation of memories at attractor states, subjects were significantly more likely than chance to report common colors, even on the half of trials when the target was drawn from a uniform distribution (Supplementary Fig. 7a, $p < 0.001$, randomization test). Third, over the course of the experiment, the pattern of bias around common colors became more consistent with attractor states. As shown in Fig. 4c, attractors pull in nearby memories, resulting in a positive-to-negative transition in bias. The more negative the slope, the stronger the attractor. Attraction towards common colors increased with experience: the slope of bias around common colors was significantly more negative during the last third of trials than during the first third (Fig. 7e, $p = 0.0138$, bootstrap).

To determine if changes in bias were driven by differences in encoding or memory dynamics, we analyzed short memory delay and long memory delay trials separately. If learned biases toward common colors manifest during encoding, then the bias slope should become more negative for both short and long trials. In contrast, if biases manifest during memory, then the change in

bias should be especially strong for long delay trials because the biases in memory dynamics have more time to accumulate. Non-parametric regression revealed a main effect of delay length on bias slope ($p = 0.026$) modulated by a delay x epoch (first or last third of trials) interaction ($p = 0.039$). The bias slope around common colors on short delay trials did not differ between the first third and last third of trials (Fig. 7f, $p = 0.384$, bootstrap) but became significantly more negative for long-delay trials ($p = 0.006$, bootstrap). Directly comparing the two delay conditions, bias slope was more negative for long-delay trials than short delay trials in the last third of trials ($p = 0.0411$, bootstrap). These results suggest that learning modified dynamics during memory rather than encoding.

To ensure that these results were not due to subjects strategically reporting common colors based on explicit knowledge of the stimulus distribution, we analyzed debriefing data collected from the participants in Experiment 2 and 1b. Subjects were not better than chance at identifying whether they were exposed to a biased or uniform stimulus distribution (see Methods for details). Furthermore, participants in Experiment 2 displayed the same pattern of results regardless of whether or not they correctly reported that the stimulus distribution was biased during debriefing (Supplementary Fig. 8).

Finally, if attractors emerge at common color locations, then this should alter the distribution of reported colors over the course of the experiment. Indeed, we found the clustering of memory reports across subjects decreased from the first third to the last third of trials (2.62 to 2.63 bits, $p < 0.001$, randomization test; Supplementary Fig. 7b). This is consistent with a strengthening of attractors at the contextually-predicted locations, which were uncorrelated across subjects. However, it is important to note that, although weaker, clustering is still partially evident at baseline locations in the last third of trials (Supplementary Fig. 7b), and the slope of bias around these baseline locations did not change in strength between the first and last third of the experiment ($p = 0.5701$, bootstrap). This suggests that the learning rate governing changes in the dynamics is low, ideal for extracting statistical regularities[31].

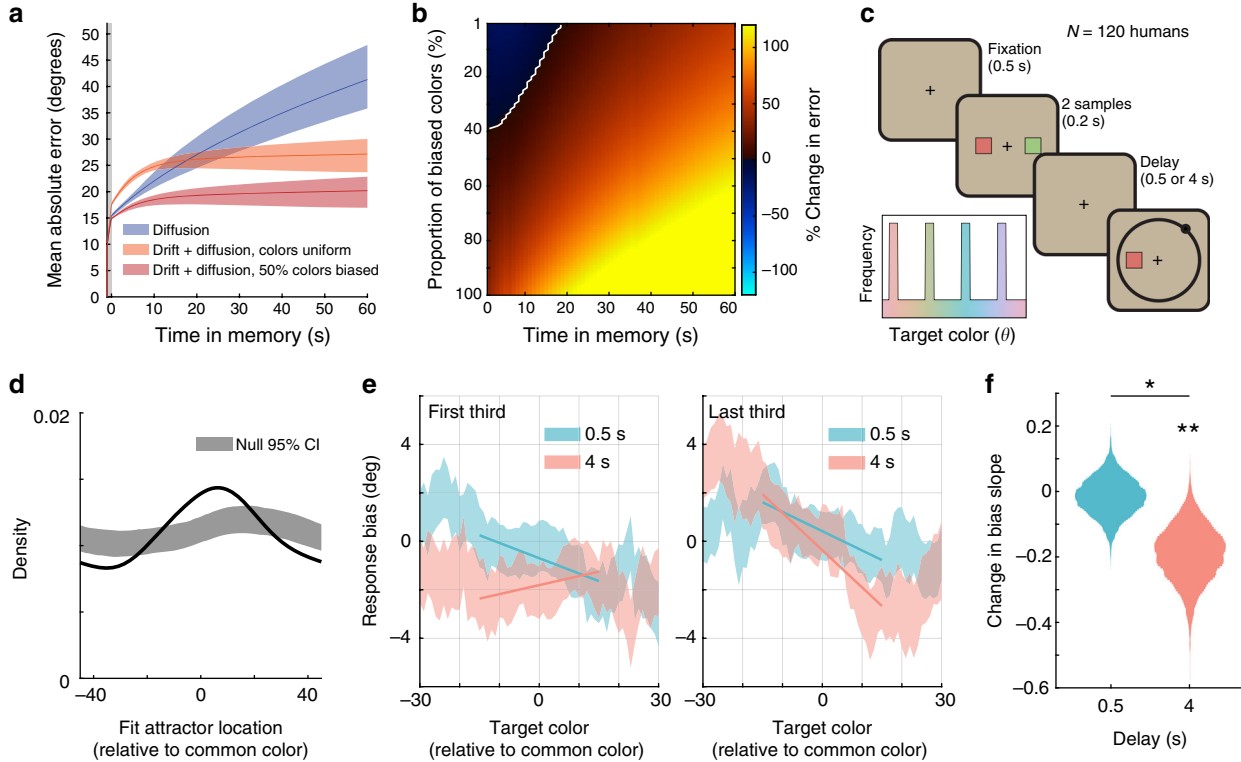

**Fig. 7** Attractor dynamics adapt to the context. **a** Simulated mean absolute error in the representation of the target color over time. The best-fitting discrete attractor model in Experiment 1a ('drift + diffusion'; orange and red) is compared to the same model without encoding or memory drift ('diffusion'; blue). Error is simulated for two target distributions: one in which targets were sampled uniformly (colors uniform) and one in which 50% of targets were drawn more frequently from colors near attractors. Note that the performance of the diffusion model does not depend on the distribution of target colors. Results are based on the mean parameter fits for the high load condition; similar results were observed for low load. Shaded regions reflect variability across subjects (95% CI). **b** Percent increase in simulated error for diffusion model compared to full 'drift + diffusion' model as a function of time in memory (x-axis) and the degree of bias in the target distribution (y-axis). Warm colors indicate attractors reduce memory error; white line indicates no change. **c** Experiment 2. Humans ($N = 120$) performed a color delayed estimation task with two stimuli. Inset: Example color distribution for one subject. Four groups of colors were presented more frequently. Common colors were equally spaced in color space and differed for each subject. **d** Probability of attractor position (estimated from dynamical model fits) relative to common colors (black). Shaded region indicates 95% confidence intervals of the null distribution, based on randomly permuting the location of common colors across subjects. **e** Bias for targets around common colors during the first and last third of trials, with regression lines. Error bars reflect standard error of the mean. **f** Difference in the slope of bias at common colors between the first third and last third of trials for short and long-delay trials. Distributions reflect bootstrapped values. *$p < 0.05$, **$p < 0.01$, bootstrap. Source data are provided as a Source Data file

## Discussion

Our results highlight the dynamic nature of working memory representations. Using both model-based and model-free analyses, we show that two forces drive the evolution of visual representations during encoding and maintenance: (1) random diffusion and (2) drift towards discrete attractor states. Together, these forces provide a parsimonious explanation of the distribution, bias, and precision of memory reports and the accumulation of error in memory over time. These results build on previous models that do not explain why errors in working memory differ as a function of the content (e.g., refs. [9,10,14]) or how memory representations dynamically evolve (e.g., ref. [24]).

Previous psychophysical, theoretical, and neurophysiological work has shown noise in neural activity can cause memories to diffuse away from their original representation, leading to errors in working memory[12–15]. Our results suggest attractor dynamics within mnemonic space can counteract this noise by pulling memories towards a few stable representations. Consistent with previous theoretical work[17–21], we provide experimental evidence that the stability of representations at attractors limits the effect of random diffusion. Furthermore, the fact that discrete attractors are evolutionarily conserved across monkeys and humans

emphasizes the benefits of error-correction. Indeed, this may be a general phenomenon in the brain: attractor dynamics are thought to minimize the impact of noise in long-term, associative memory[32,33] and in decision making[34,35].

From an information-theoretic perspective, working memory can be conceptualized as a band-limited information channel[36]. In this framework, discrete attractors compress working memory representations by discretizing the continuous mnemonic (color) space. Discretization reduces the information needed to encode a memory, allowing it to be more accurately stored in a noisy, band-limited system[36,37]. This is particularly important when storing multiple items in working memory. Increasing the number of items in working memory leads to interference between items, reducing memory accuracy[4,27,38]. Consistent with this, we observed an increase in diffusive noise as more items are held in working memory. However, drift also increased in strength, compensating for the increase in noise. In other words, strengthening discrete attractor dynamics increases compression of memories; this reduces the fidelity of memories as they are further discretized, but also makes them more robust to noise and interference. Note that this increase in attractor strength with load cannot be explained by interference among items because

item identity is random and so any such interactions would lead to random, not systematic, biases in memory. Several neural mechanisms might account for the increase in attractor strength with load, including increased drive into the network[39,40] or changes in f-I gain via neuromodulation[41,42].

Finally, our results suggest attractor dynamics adapt to context: attractors emerged at the position of commonly occurring stimuli. The relatively slow rate of change in dynamics (over hundreds of trials) is consistent with theoretical work that suggests such learning could be driven by synaptic plasticity[43]. Indeed, such a mechanism with a slow learning rate is ideal for extracting the statistical regularities of the environment. Intriguingly, we found encoding dynamics adapted to changes in the environment more slowly than memory dynamics. This raises the possibility that encoding and memory dynamics may rely on different neural mechanisms.

By moving to reflect the statistics of the environment, attractors will pull memories towards likely stimuli. In this way, attractor dynamics act to integrate prior beliefs with noisy stimulus information. This process is analogous to Bayesian inference applied over time. At each timestep in memory, drift applies the prior (embedded in the attractors) to each item in memory, which reflects the posterior of the previous timestep plus random noise. Thus, as time in working memory increases (and stimulus information diffuses), memory representations drift towards prior expectations. Such a process could constitute the mechanism by which sensory history influences working memory[44–46]. Beyond working memory, attractor dynamics could be a neurally-plausible mechanism for integrating prior beliefs with sensory information in other cognitive behaviors, such as decision making and perception.

## Methods

**Participants**. Thirty-three human subjects participated in Experiment 1a at Princeton University. Seventy-three additional subjects participated in an online version of Experiment 1a via Amazon Mechanical Turk (https://www.mturk.com). One-hundred twenty-five subjects participated in Experiment 1b via Amazon Mechanical Turk. One-hundred fifty-five subjects participated in Experiment 2 via Amazon Mechanical Turk. We screened subjects for a minimum of engagement in the task by estimating their probability of random guessing in the task using 3-component mixture model[8]. Subjects with an estimated guess rate greater than 20% across all trials were excluded from further analysis, yielding thirty laboratory subjects and sixty online subjects for Experiment 1a, one-hundred twenty online subjects for Experiment 1b, and one-hundred twenty online subjects for Experiment 2. This threshold of 20% was set independently based on analysis of a separate pilot cohort of online subjects ($N = 57$). Subjects recruited online via Mechanical Turk have previously been used to study working memory and have performance comparable to lab subjects[47,48]. We observe similar qualitative behavior between online and lab subjects (Supplementary Fig. 9) and report their behavior together in the main text. All subjects attested that they had normal or corrected-to-normal vision. We confirmed that subjects had normal color vision using the Ishihara Color Blindness Test. Subjects provided informed consent in accordance with the Princeton University Institutional Review Board.

Two adult male rhesus macaques (8.9 and 12.1 kg) performed the Experiment 1a in accordance with the policies and procedures of the Princeton University Institutional Animal Care and Use Committee.

**Experiment 1a - humans**. For the laboratory version of Experiment 1a we presented stimuli on a CRT monitor positioned at a viewing distance of 60 cm. We calibrated the monitor using an X-Rite i1Display Pro colorimeter to ensure accurate color rendering. During the experiment, participants were asked to remember the color and spatial location of either 1 or 3 square sample stimuli. The color of each sample was drawn from 360 evenly spaced points along an isoluminant circle in CIELAB color space. This circle was centered at ($L = 60$, $a = 22$, $b = 14$) and the radius was 52 units. Colors were drawn pseudorandomly, with the caveat that colors presented on the same trial had to be at least 22° apart in color space. The samples measured 2° of visual angle (DVA) on each side. Each sample could appear at one of eight possible spatial locations. All possible locations had an eccentricity of 4.5 DVA and were positioned at equally spaced angles relative to central fixation (0, 45, 90, 135, and 180° clockwise and counterclockwise relative to the vertical meridian). The dimensions of the stimuli for the online experiment were defined by pixels rather than degrees of of visual angle. The samples had an edge length of 30 pixels and were presented at an eccentricity of 170 pixels.

Participants initiated each trial by clicking the mouse and by fixating a cross at the center of the screen (Fig. 1a). After 500 ms of fixation, one or three samples (the load) appeared on the screen. The samples were displayed for 200 ms and then were removed from the screen. Participants then experienced a memory delay of 1 or 7 s, after which a response screen appeared. The response screen consisted of the outline of a square at one of the previous sample locations (the probe sample) and a response interface consisting of a circle on a ring. Participants used the mouse to drag the circle around the ring, which changed the color of the probe sample. The angular position of the circle on the ring corresponded to a particular angle in color space. The mapping between circle position and color space was randomly rotated on each trial to exclude the use of spatial memory. We instructed participants to adjust the color of the probe sample to match the color of the sample that had previously appeared at that location as closely as possible. We told participants that accuracy was more important than speed but that they should respond within a few seconds. There was no time limit on the response. All human participants completed 200 trials.

We monitored the eye position of the lab participants using an Eyelink 1000 Plus eyetracking system (SR Research). Participants had to maintain their gaze within a 2° circle around the central cross during initial fixation and sample presentation, or else the trial was aborted and excluded from analysis.

**Experiment 1a - monkeys**. We presented stimuli on a Dell U2413 LCD monitor optimized for color rendering. The monitor was positioned at a viewing distance of 58 cm. We calibrated the monitor using an X-Rite i1Display Pro colorimeter to ensure accurate color rendering. Sample colors were drawn from 64 evenly spaced points along an isoluminant circle in CIELAB color space. This circle was centered at ($L = 60$, $a = 6$, $b = 14$) and the radius was 57 units. Slightly different color wheels were used for the humans and the monkeys to accommodate the gamut of the different monitors used in each experiment. Nevertheless, colors corresponding to the same angle in each color wheel are extremely similar in appearance. The edges of the samples measured 2° of visual angle. Each sample could appear at one of two possible spatial locations: at 5 DVA eccentricity from fixation and 45° clockwise and counterclockwise from the horizontal meridian.

We adapted Experiment 1a so that it could be performed by non-human primates. The animals initiated each trial by fixating a cross at the center of the screen. After 500 ms of fixation, one or two samples appeared on the screen. The samples were displayed for 500 ms, followed by a memory delay of 500 or 1,500 ms. Next, a symbolic cue was presented at fixation for 300 ms. This cue indicated which sample (top or bottom) the animal should report in order to get juice reward. The response screen consisted of a ring 2° thick with an outer radius of 5°. The animals made their response by breaking fixation and saccading to the section of the color wheel corresponding to their report. This ring was randomly rotated on each trial to prevent motor planning or spatial encoding of memories. The animals received a graded juice reward that depended on the accuracy of their response. The number of drops of juice awarded for a response was determined according a circular normal (von mises) distribution centered at 0° error with a standard deviation of 22°. This distribution was scaled to have a peak amplitude of 12, and non-integer values were rounded up. When response error was greater than 60°, no juice was awarded and the animal experienced a short time-out of 1 to 2 s. Responses had to be made within 8 s; in practice, this restriction was unnecessary as response times were on the order of 200–300 ms. We analyzed all completed trials (trials on which the animal successfully maintained fixation and saccaded to the color wheel, regardless of accuracy). Monkey W completed 15,787 trials over 26 sessions and Monkey E completed 16,601 trials over 17 sessions.

We monitored the eye position of the animals using an Eyelink 1000 Plus eyetracking system (SR Research). The animals had to maintain their gaze within a 2° circle around the central cross during the entire trial until the response, or else the trial was aborted and the animal received a brief timeout. Trials during which the animal broke fixation were excluded from analysis.

**Experiment 1b**. The stimuli and procedures were similar to those for the online version of Experiment 1a, except that participants were presented with two samples on every trial and the delay varied continuously between 1 and 7 s. Model predictions (Supplementary Fig. 6) were generated from the best-fitting model. As in Experiment 1a, the full model provided the best fit to the data (mean increase in cross-validated log-likelihood over worst-fitting model, full: 7.45, $\beta = 0$: 7.41, $\beta^* = 0$: 0.20).

**Experiment 2**. The stimuli and procedures for Experiment 2 (Fig. 7c) were similar to those for the online version of Experiment 1a. We shortened the memory delays to 500 ms and 4000 ms to reduce the length of the experiment. Participants saw 2 samples on each trial. Critically, the color of the samples were no longer always drawn uniformly from the circular color space. Rather, for each sample, there was a 50% chance that the color of that sample would be drawn from a biased distribution (Fig. 7c). This biased distribution consisted of four equally spaced clusters of common colors. Each cluster was 20° in width. Each participant was exposed to a unique set of common colors as the cluster means were shifted by a single random phase for each subject.

**Subject Debriefing**. Participants in Experiments 1b and 2 were presented with following debriefing question: "During this experiment, some participants are shown target colors at random. Others are shown some colors more often than others. Which group do you think you are in?". The response options were "I was shown all colors about equally often" or "I was shown some colors more often than others". When presented with this two-alternative forced choice at the end of the experiment, 49.2% of participants in Experiment 2 correctly reported that the distribution of targets was biased, while 48.3% incorrectly reported a uniform distribution of targets (3 participants abstained). We estimated the false alarm rate for this question by analyzing responses of participants in Experiment 1b to the same question: 49.2% incorrectly reported a biased distribution, 50.0% reported a uniform distribution, 1 abstained. The proportion of subjects reporting a biased distribution was not significantly different between Experiments 1b and 2 ($\chi^2(1) = 0.015$, $p = 0.902$, $\chi^2$).

**Effects of load and time on mean error**. Throughout the text, all t-tests are two-tailed and all randomization tests are one-tailed, unless otherwise indicated.

We analyzed mean absolute error for human subjects using a $2 \times 2$ repeated measures ANOVA with factors load, delay time, and their interaction. We analyzed each monkey's data by fitting the equivalent regression model to their mean error in each condition. We obtained bootstrapped confidence intervals for each regression coefficient by resampling trials with replacement from each monkey's dataset and refitting the regression model on each iteration (1,000 iterations). We also used this method to analyze the effect of load and time on clustering and mean bias (Supplementary Fig. 1), and the effect of task epoch and time on bias slope (Fig. 7e-f).

**Clustering Metric**. We observed that the distribution of reported hues $\hat{\theta}$ are clustered relative to the distribution of target hues $\Theta$. To quantify this phenomenon, we developed a simple clustering metric. This metric relies on the fact that entropy is maximized for uniform probability distributions. In contrast, probability distributions with prominent peaks will have lower entropy. Because the target hues are drawn from a circular uniform distribution, the entropy of the targets $H$ ($\Theta$) will be relatively high. If a subject's responses are clustered, their entropy $H(\hat{\theta})$ will be relatively low. Taking the difference of these two values yields a clustering metric $C$. Negative values of $C$ suggest greater clustering:

$$C = H(\hat{\theta}) - H(\Theta), \tag{1}$$

where:

$$H(x) = -\sum_{x=1}^{360} f(x)\log_2 f(x)\,d\hat{x}. \tag{2}$$

To account for the fact that this estimate of entropy is biased, we subsampled the data such that there was an equal number of trials in each condition. We estimated the pdf of the responses $f(\hat{\theta})$ and the targets $f(\Theta)$ using kernel density estimation (Matlab CircStat toolbox, kernel width = 10°). Note that our goal was to quantify the clustering of reports for items in memory; random guesses[7,8] confound this analysis by contributing a uniform component to the response distribution that varies systematically as a function of load and time. To address this, we estimated the proportion of responses due to guessing using mixture models[7,8] and removed a uniform component from the response distribution $f(\hat{\theta})$ and the target distribution $f(\theta)$ equal in area to the guess rate and then renormalized each.

**Bias and standard deviation of memory reports**. To dissociate systematic and unsystematic sources of error in memory, we calculated the bias and standard deviation of memory reports across color space. We used 4° bins for humans and 6° bins for monkeys to accommodate their coarser sampling of color space (64 target colors). Bias refers to the distance between the the target color and the mean reported color. We calculated the slope of bias around negative-slope zero-crossings in each subject's fit drift function (Experiment 1a), around significant peaks in each subject's response histograms (Experiment 1a), and around commonly presented presented colors (Experiment 2) by fitting a line to the bias $+/-15°$ around the point of interest. Mean standard deviation around these points was calculated around these points using the same window ($+/-15°$). For monkey subjects, we boostrapped confidence intervals for slope and standard deviation by resampling trials with replacement.

To compute the bias and SD for the non-uniform guessing strategy (Supplementary Fig. 4), we performed 1,000 iterations of a randomization test where memory reports were shuffled with respect to the target colors and report the mean bias and SD for each target color across iterations.

To identify significant peaks in subjects' response histograms (Experiment 1a), we first estimated the PDF of subjects' responses using kernel density estimation. We identified possible peaks as samples larger than their two neighboring samples and recorded their amplitude. We then repeated this analysis on the distribution of targets, resampling with replacement to create a null distribution of peak amplitudes. Peaks in the original response distribution with an amplitude greater than the 95th percentile relative to the null were deemed significant. We identified

negative-slope zero-crossings in the fit drift function of each subject by identifying peaks in the numerical integral of the drift function. Peaks with a prominence in the 20th percentile or lower across subjects were excluded from analysis.

Finally, to generate model predictions for bias and standard deviation, we fit the discrete attractor model to each subject's data and generated synthetic datasets (1,000 trials for each human subject and 20,000 trials for each monkey) by simulating responses from each subject's best-fit model. We then analyzed the bias and standard deviation of these simulated reports as above. Model performance was assessed by correlating model predictions with empirical results across target colors.

**Dynamical Model**. We developed a quantitative model to describe how items in memory change over time. We assume that two distinct influences may make memory dynamic. First, systematic biases may cause memories to drift towards stable attractor states over time. Second, memories may be perturbed by unsystematic random noise. We model memory using a stochastic ordinary differential equation that captures both of these influences:

$$d\theta = \beta_L G(\theta)dt + \sigma_L dW. \tag{3}$$

This equation describes the time evolution of a color memory $\theta$ (a circular variable corresponding to an angle in our circular color space) under the influence of some deterministic dynamics defined by $G$ (the drift) as well as an additive white noise process $W$ with variance $\sigma^2$. $\beta_L$ sets the gain of the drift. Thus, $\beta_L G(\theta)dt$ describes influence of drift and $\sigma_L dW$ the influence of random noise on memory. To test the hypothesis that memory load influences these dynamics we fit a separate $\beta$ and $\sigma$ for each load $L$.

Based on the clustering we observe in the data, it seems likely that $G(\theta)$ is a nonlinear function. We needed a relatively parsimonious way of describing $G(\theta)$ that still gave us enough flexibility to describe this nonlinearity. So, for each subject, we defined $G(\theta)$ using a basis set consisting of twelve first derivatives of the von mises distribution separated by 1 standard deviation on the interval $(0, 2\pi)$:

$$G(\theta) = \sum_{j=1}^{12} w_j \frac{d}{d\theta}\phi\left(\frac{2\pi}{12}j, \frac{2\pi}{12}\right), \tag{4}$$

where $\Phi$ is a von mises distribution parameterized by a mean and standard deviation. We then divided $G(\theta)$ by its maximum absolute value. This normalization procedure aids the interpretation of $\beta$: it is the maximum instantaneous drift rate. Our choice of 12 basis functions was to minimize AIC in comparison to function estimates with a higher or lower number of basis functions.

To fit the model described in Eq. (3) to subject data, we needed to describe the time evolution of $\theta$ probabilistically. So, we rewrote Eq. (3) as a Fokker-Planck equation, a partial differential equation that tracks the probability density function of $\theta$ over time:

$$\frac{\partial}{\partial t}p(\theta, t) = -\frac{\partial}{\partial \theta}\beta_L G(\theta)p(\theta, t) + \frac{\sigma_L^2}{2}\frac{\partial^2}{\partial\theta^2}p(\theta, t). \tag{5}$$

In order to track probability mass, we discretized our 1-dimensional state space (the value of $\theta$) into 100 evenly spaced bins from 1° to 360°. Once discretized, the change in $p(\theta, t)$ over a given timestep $dt$ can be described by a Markov transition matrix $M_L$:

$$\frac{\partial}{\partial t}p(\theta, t) = M_L p(\theta, t). \tag{6}$$

This discretized approximation can be solved analytically in time, yielding:

$$p(\theta, t) = e^{M_L t}p(\theta, 0), \tag{7}$$

where $p(\theta, 0)$ is the initial state of memory after encoding.

We wanted to dissociate load-driven changes in the dynamics of memory and encoding. To capture differences in encoding, we allowed the state of a memory at the start of the delay, $p(\theta, 0)$, to vary as a function of load. To simulate the encoding process, we first initialized a narrow probability density $P_0(\Theta)$ that reflects the color of the target stimulus. $P_0$ is a von mises distribution with mean equal to the target color $\Theta$ and a standard deviation of 0.1 radians:

$$P_0(\Theta) = \phi(\Theta, 0.1). \tag{8}$$

We then allowed $P_0$ to propagate for a 1 s encoding period according to the following differential equation:

$$d\theta = \beta_L^* G(\theta)dt + \sigma_L^* dW, \tag{9}$$

where $\beta_L^*$ and $\sigma_L^*$ interact to set the bias and variance of the encoded memory. Therefore, $p(\theta, 0)$ is calculated as:

$$p(\theta, 0) = e^{M_L^*}P_0(\Theta) \tag{10}$$

and the final probability distribution describing the memory of the target hue after a memory delay of $t$ seconds on a trial with load $L$ is:

$$p(\theta, t) = e^{M_L t}e^{M_L^*}P_0(\Theta). \tag{11}$$

All drift and diffusion parameters ($\beta_L$, $\sigma_L$, $\beta_L^*$ and $\sigma_L^*$) are rates; they measure the change in memory over time (either due to drift or diffusion). However, care must be taken when directly comparing the value of these parameters across the

encoding and memory periods. This is because encoding is modeled as occurring over a fixed period (1 s), while the length of the memory delay can change from trial to trial. Therefore, the degree to which memory dynamics influence reports depends on the length of the memory delay. Drift and diffusion can be compared more directly within the encoding or memory periods.

Equation 11 describes the probability distribution for the memory of the target color $\Theta$ at time $t$. However, our goal is to predict the subject's report on a particular trial, $p(\hat{\theta}, t)$, which does not just depend on the color of the target[7,8]. On some trials, subjects may experience complete failures of memory, resulting in random guessing. On other trials, subjects may commit a 'swap' error and report their memory of one of the non-target colors, $\theta_i^*$ (note that the memory of non-target colors also evolved according to Eq. 11). Finally, random error may be introduced at decoding. To account for these additional influences, we estimated each subject's probability of committing swap errors and guessing, and, for each trial, computed a mixture of the target memory distribution, the non-target memory distributions, and a uniform component:

$$p(\hat{\theta}, t) = (1 - \lambda - \alpha)p(\theta, t) + \alpha \frac{1}{m} \sum_{i=1}^{m} p(\theta_i^*, t) + \lambda \frac{1}{2\pi}, \quad (12)$$

where $m$ is the number of non-target colors (0 or 2 for humans, 0 or 1 for monkeys). $\alpha$ and $\lambda$ represent the probability of swap errors and guesses, respectively. They are linear functions of $t$ parameterized by a slope $a$ and intercept $b$. We estimated a unique $\lambda$ and $\alpha$ function for each load (note that $\alpha$ takes on a value of zero when load is 1). To capture decoding error, we circularly convolved the final response distribution with a von mises distribution with a standard deviation $\sigma^{\dagger}$. As noted below, we found the model with response error fit well to human behavior. However, monkey behavior was best explained without this term.

We found the maximum likelihood estimate (joint likelihood across trials) of the free parameters $\beta_L$, $\beta_L^*$, $\sigma_L$, $\sigma_L^*$, $a_{\lambda_L}$, $b_{\lambda_L}$, $a_\alpha$, $b_\alpha$, $w_j$, and $\sigma^{\dagger}$ (humans only) using gradient descent. To obtained boostrapped distributions of the parameter distributions for human subjects, we repeatedly resampled the parameters fit to each subject with replacement and took the mean of these values. To obtain bootstrapped distributions for monkey subjects, we repeatedly resampled each monkey's pool of trials with replacement and repeated the fitting process. Model comparison was performed on data pooled across sessions (monkeys) or subjects (humans).

Model fits indicated that random guessing increased with time for human subjects (Supplementary Fig. 10), consistent with previous reports[2–4]. Guessing decreased with delay, however, for the two monkeys. We wanted to ensure that trade-offs between guessing and other parameters, such as the rate of diffusion, were not driving the effects of increased drift and diffusion with load. So, we fit different versions of the model in which we systematically simplified our parameterization of guess rate. Across the two monkeys, model comparison using AIC and BIC indicated that the full model was the best fit to the data. Regardless, for all models, drift and diffusion increased with load, indicating that this is a stable feature (Table S2).

Model comparison indicated that the full model including decoding error was clearly better than the model without decoding error in humans. However, the model with decoding error was not clearly better than a model without decoding error across monkeys and so we defaulted to the simpler model (monkey E: $wBIC = 0.01$; monkey W: $wBIC = 1.00$; compared to $wBIC = 1.00$ in humans). Furthermore, in exploratory tests we found decoding error substantially disrupted the ability of the model to predict the clustering and precision of responses in monkey W; with decoding error the correlation between the predicted and observed response distribution in monkey W dropped from .741 to .393 and the correlation between the predicted and observed pattern of precision across colorspace dropped from .377 to .120. Based on this, we concluded that models with decoding error best described the human behavior but that the simpler model without decoding error best described the monkey behavior. Differences in decoding error could reflect different response modalities (moving a mouse for humans, saccade for monkeys) or reflect the fact that monkeys saw the entire color wheel while humans did not.

**Simulated error of models over time**. We wanted to identify if attractor dynamics might be normative and enhance the fidelity of memory. To do this, we computed the expected mean error for the memory of a target color as a function of delay time for the full dynamic model with attractor dynamics (drift + diffusion) and a model without attractor dynamics (diffusion). The drift and diffusion parameters of the drift + diffusion model were set to the mean fit parameters for the human subjects in Experiment 1a. The parameters of the diffusion model were identical except that $\beta_L$ and $\beta_L^*$ were set to zero. To isolate error in the representation of the target color, the probabilities of guessing and swaps were set to zero. To create a representative drift function, we fit our basis set to the numerical derivative of the PDF of the response distribution for human subjects (normalized to have a maximum absolute value of one), which yields attractors at locations in color space where they are most frequently observed (i.e., at commonly reported colors). To create biased target distributions, we parametrically took a weighted average of a distribution that was entirely uniform over color space and a biased distribution that was uniformly distributed within 10 degrees of attractor states and zero elsewhere.

**Nonlinear mapping between stimulus and perceptual space**. The color space used to parameterize stimuli in these experiments (CIELAB) is designed to be perceptually uniform, but we sought to demonstrate that inhomogeneties in this space cannot explain our results. To demonstrate this, we analyzed an alternative model (Supplementary Fig. 5a) which assumes a nonlinear mapping between our stimulus parameterization (a circle in CIELAB space) and a hypothetical true perceptual space (a square, although the results generalize to other shapes). The continuous CIELAB and perceptual spaces were discretized into 1,024 points. We simulated memory reports by first generating 100,000 angles randomly distributed around our stimulus space (representing the target stimuli) and projecting these points onto the true perceptual space (representing encoding). Memory was simulated as a purely diffusive process of the encoded target colors around the true perceptual space (i.e., there were no discrete attractor dynamics). Simulations were run for 1,000 timesteps (arbitrary units). Diffusive noise at each timestep was modeled as random step between 0 and 4 points in either direction in the discretized perceptual space. Report was simulated by projecting the diffused memory representations back into stimulus space. This model predicts clustering of memory reports (Supplementary Fig. 5a) but does not predict attractive bias around cluster peaks (Supplementary Fig. 5b) as observed empirically. We thank and anonymous reviewer for proposing and implementing this alternative model.

**Reporting summary**. Further information on research design is available in the Nature Research Reporting Summary linked to this article.

## Data Availability
All data that support the findings of this study are available from the corresponding author upon request. The source data underlying all figures and Supplemental Tables 1 and 2 are provided as a Source Data file.

## Code Availability
Code for fitting the discrete attractor model to behavioral data from delayed continuous report tasks is available at https://github.com/buschman-lab.

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

## Acknowledgements

We thank A. Piet for suggesting trial-by-trial analysis, B. Morea and H. Weinberg-Wolf for assistance with NHPs, and S. Henrickson, F. Bouchacourt, A. Libby, and P. Kollias for comments. This work was supported by NIMH R56MH115042 and ONR N000141410681 to TJB, an NDSEG fellowship to MFP, and McKnight Foundation, Simons Collaboration on the Global Brain (SCGB AWD1004351) and the NSF CAREER Award (IIS-1150186) to J.W.P.

## Author contributions

M.F.P. and T.J.B. conceived of the experiments; M.F.P., B.D., J.W.P. and T.J.B. designed the dynamical model; M.F.P. and B.D. implemented the model; M.F.P. collected and analyzed the data; M.F.P. and T.J.B. wrote the original draft; M.F.P., B.D., J.W.P. and T.J.B. discussed the results and prepared the final draft.

## Additional information

**Competing interests:** The authors declare no competing interests.

