## [Peer Review File · Nature Communications]

Reviewers' Comments:

Reviewer #3:

Remarks to the Author:

Apologies for the delay in getting my comments back to you. As noted in the first round, this is a really nice paper. The additional explanations and data that the authors added now make it suitable for publication in Nat Comm, and I think that the authors did a nice job replying to all of the reviewers. Great job and congratulations!

Reviewer #4:

Remarks to the Author:

The manuscript has improved since the last review, but it is still unsatisfactory for the following reasons:

A major concern is still the advance over previous results in the literature. Some of the results of this study were previously reported by Bae et al. 2014, 2015, and this is still not properly disclosed in the text. A specific paragraph stating what we know about color biases with and without delay is necessary either in introduction or as the analyses are shown. The text still does not convey clearly what is known and what are new results. In particular, figure 4, now presented as one main result, replicates results from Bae et al. 2015, Figs. 7-8.

The manuscript should not promote the wrong view that report biases are mostly the result of memory drift. Authors are reluctant to compare quantitatively the contribution of encoding and memory processes to report biases, and they argue that model parameters cannot be compared. This is unsatisfactory. The argument about parameter rates not being comparable is weak, and the authors are surely able of finding reasonable measures for comparison. In the previous studies (Bae et al. 2014, 2015), this had been established by using zero delay conditions, which is the most direct test. Alternatively, their new cross-validation analysis in table S2 also quantifies how important each drift component is to describe the behavioral data.

line 72: ".. suggesting that clustering is the result of a load-dependent dynamic process that unfolds over the course of the memory delay and not simply introduced at encoding" is ambiguous in this respect, possibly meaning that encoding did not introduce biases.

There is still circularity in some of the main arguments, which has not been clarified in the revised manuscript. Figure 4 is presented as support for discrete dynamics, but the data reported in Figure 4 is mostly just validating the fit. Any possible fit of the data that includes the possibility of non-homogeneous reports will yield similar results. In particular, both the fact that reports cluster around fitted "attractors" (Fig. 4a) and that reports are less disperse around these "attractors" (Fig. 4c) are a simple byproduct of having fitted correctly the nonhomogenous set of reports with just any reasonable model. See the attached code gaussianfit.py or the model used by Bae et al. 2014, 2015. Authors responded to this criticism by mentioning an additional cross-validation approach, but this does not address my concern, it just shows that systematic biases are true biases and not due to random

fluctuations, but not that the discrete dynamics model is particularly compelling. The one piece of data in this figure that does support the discrete dynamics model is the bias (Fig. 4c,d). This is the main point and needs to be specifically stressed. As shown (and nicely interpreted) in their rebuttal, the NHR model predicts a different pattern of bias, and so does a modulated Gaussian model (gaussianfit.py). In my view, Fig. 4a,b,e,f belong more to Fig.2, where the model is presented and it is shown that it can fit the data (as many other models, previously shown by Bae et al. 2014, 2015). Instead, fig. 4 would be more about presenting the bias as the critical feature to look at to discriminate between alternative models, something it has not been shown before.

The new data with parametric variation of delay duration is very nice. However, the reporting of results should be improved. The model fit is now relegated to Fig. S6 but this is not in line with the main point of the manuscript, that seeks to establish the plausibility of a specific model. The model fit should be in the main figure, and its deviation from a straight line should be established statistically instead of resorting to a power law fit that lacks any specific motivation. Parameters for this model fit should also be reported. Also, there is an apparent discrepancy between the power law fit now reported in Fig. 5d and the (quite linear) model fit in Fig. S6. Is it correct to interpret that the data is consistent with a set of very shallow discrete attractors, approximating imperfectly a line attractor?

The simulation results in Fig. 7b are surprising. It seems like the "drift+diffusion" model (the full model of fig. 3a) has very similar average mean error at the beginning of the memory period than the diffusion model (with neither encoding nor memory drift, i.e. similar to $\beta^*=0$ in fig. 3a when considering time $t=0$). How is that possible? Since the encoding period has the largest impact on drift (Table S2, Fig 6c,d), one would intuitively expect the "drift+diffusion" model to have a much larger mean error at the beginning of memory than the diffusion model. This can be clarified by showing in fig. 7a-b the simulation data from the encoding period, too (as in fig. 3a). Also, there is no mention of how more frequent "attractor" colors have to be in order for the "colors biased" condition to make the drift+diffusion model better than the diffusion model at all times.

The new analysis with short and long delay trials is very interesting. The comparisons between short/long and first/last third should be supported with an ANOVA-like analysis. Surprisingly, for short delay trials there appears not to be much change between the last third and the first third of the session. This suggests that the "context adaptation" mechanism affects only the memory drift component, and not the encoding drift component of the model. Based on this result, memory and encoding drifts will typically respond to different sets of attractors (encoding drift will be fixed, while memory drift will adapt). How does this influence the reasoning in Fig. 7b? Intuitively, the "colors biased" condition will be adapted to the memory drift, but not to the encoding drift, so there will always be a behavioral cost to using the "drift+diffusion" model for short memory delays.

The last paragraph of results concludes that clustering reduction is consistent with weakening of baseline attractors, but it could be solely the result of strengthening of uncorrelated context-defined attractors (as shown in Fig. 7). As used in Fig. 7, the real test for strengthening or weakening of attractors is the response bias at the location of the attractor. It is not justified not to report this direct test of the modulation of baseline attractor strength. The prediction to be tested is that bias slopes around baseline attractors should get less negative in the last third compared to the first third of the session.

Minor:

- model parameters sometimes have an n subindex, sometimes an L subindex.
- Eq. (12): α should vanish for load 1 trials. The current notation does not reflect that.
- Fig. R6 is incorrect. The NHR model does not show a linear increase, as readily computed with

simulations. Also, the discrete attractor model curve is far from the data fit (Fig. S6). In practice, these two models are indistinguishable in this graph.

- authors resolve one problem in interpretation of Fig. 7 by resorting to debriefing data from participants. The specific question that participants responded to should be literally stated in Methods.

```

import numpy as np
import matplotlib.pyplot as plt
from numpy.random import normal
import scipy.stats as stt

# Gaussian fit model
n_trials = 10000000
nbins = 50

targets = np.random.random(n_trials)*2.*np.pi
sigma = 0.3*(1.5+np.cos(4.*targets))/2.5 #this is the G(theta) function
that should be fitted to data, for now we assume it is a cosine function
with 4 maxima. The gain factor 0.3 could in addition include dynamics. As
this gain factor increases, clustering increases.
reports = targets + normal(0,sigma,n_trials)
reports[reports>2*np.pi] = reports[reports>2*np.pi]-2*np.pi
reports[reports<0] = reports[reports<0]+2*np.pi

err = reports - targets
err[err>2*np.pi] = err[err>2*np.pi]-2*np.pi
err[err<0] = err[err<0]+2*np.pi

def ccstd(x):
    return stt.circstd(x,high=np.pi, low=-np.pi)
def ccmean(x):
    return stt.circmean(x,high=np.pi, low=-np.pi)

prec = stt.binned_statistic(targets, err, statistic=ccstd, bins=nbins)
bias = stt.binned_statistic(targets, err, statistic=ccmean, bins=nbins)
count = stt.binned_statistic(reports, reports, statistic='count',
bins=nbins)

plt.figure(figsize=(5,5))

plt.subplot(3,1,1)
plt.plot(count[1][:-1], count[0], "k")
plt.title("density of reports")
plt.xticks([])

plt.subplot(3,1,2)
plt.plot(bias[1][:-1], bias[0], "r")
plt.title("bias")

plt.subplot(3,1,3)
plt.plot(prec[1][:-1], prec[0], "r")
plt.title("std")

plt.tight_layout()
plt.show()

```

We thank the reviewer's for their feedback and enthusiasm about the manuscript. In response to the reviewers comments, we have improved the analysis and presentation of results in Figures 3-7 and have also added important clarifications to the methods and discussion. We believe these changes have strengthened the manuscript. Original comments are in bold blue, our responses are in black text.

Reviewer #3:

Apologies for the delay in getting my comments back to you. As noted in the first round, this is a really nice paper. The additional explanations and data that the authors added now make it suitable for publication in Nat Comm, and I think that the authors did a nice job replying to all of the reviewers. Great job and congratulations!

Thank you very much!

Reviewer #4:

Comment 1. A major concern is still the advance over previous results in the literature. Some of the results of this study were previously reported by Bae et al. 2014, 2015, and this is still not properly disclosed in the text. A specific paragraph stating what we know about color biases with and without delay is necessary either in introduction or as the analyses are shown. The text still does not convey clearly what is known and what are new results. In particular, figure 4, now presented as one main result, replicates results from Bae et al. 2015, Figs. 7-8.

This is a repeat of a previous comment from the reviewer. As we noted in our last reply, the goal of Figure 4 is not to show that the clustering, bias, and precision of color reports varies across color space in both humans and monkeys. The goal of Figure 4 is to show that the discrete attractor model captures these effects. In the previous round of review, we addressed this concern by revising the introduction to Figure 4 to clarify that this is our goal and to acknowledge the prior work by Bae et al. To further attempt to address this issue to the reviewer's satisfaction, we have expanded our discussion of the Bae et al papers. On line 115-117 we now state:

As seen in previous work, (Bae et al., 2014, 2015) memory reports clustered cluster at certain points in color space, and the bias and precision of reports varied systematically around points of peak clustering. Here, we show that discrete attractor dynamics explain these variations.

Additionally, when motivating the inclusion of an encoding stage in our model on line 95-97, we now state:

Previous work has shown that reports of perceived colors are clustered, although clustering is greater for colors held in working memory (Bae et al, 2015). To capture clustering and other sources of error (Buschman et al. 2011, Bays et al. 2011) that emerge during encoding, inputs were first passed through an encoding stage...

Comment 2.1 The manuscript should not promote the wrong view that report biases are mostly the result of memory drift.

We agree with the reviewer. Our intention was not to claim that report biases are mostly the result of memory drift. Instead, our results show that attractor dynamics (during encoding and memory) can explain multiple facets of behavior, including bias, clustering, and precision (Figure 4), are modulated by load (Figure 6), and have a normative explanation (Figure 7). However, it is important to note that drift during the memory delay has a significant and meaningful effect on memories. Both model-based and model-free analyses show that drift dynamics occur during the memory delay and that they impact behavior (Figure 3, Figure 5, Figure S1-2). To address this concern, we have edited the text to emphasize that attractor dynamics during both encoding and the memory delay contribute to bias (lines 73, 100-101, 113-114, and 262-265).

Authors are reluctant to compare quantitatively the contribution of encoding and memory processes to report biases, and they argue that model parameters cannot be compared. This is unsatisfactory. The argument about parameter rates not being comparable is weak, and the authors are surely able of finding reasonable measures for comparison. In the previous studies (Bae et al. 2014, 2015), this had been established by using zero delay conditions, which is the most direct test. Alternatively, their new cross-validation analysis in table S2 also quantifies how important each drift component is to describe the behavioral data.

We agree with the reviewer that the effect of encoding and maintenance processes on working memory should be clear. Indeed, this is why we constructed the model such that the encoding and memory drift terms have the same units and therefore can be compared. However, as noted in our previous response and in the previously revised manuscript, one must account for the fact that parameters are reported as rates. While the encoding parameters exert their influence over simulated fixed time period (1 second), memory parameters exert their influence for as long as an item is held in memory. As a result, even relatively small memory parameters will exert a large influence on memory as delay increases. For example, Figure 7a shows that memory drift and diffusion causes a 19.6% increase in error relative to post-encoding after a 3-second delay even though these parameters are 2-5x smaller than their encoding counterparts. We have now further updated the text to further clarify this point to the reviewer's satisfaction [line 186-187].

Second, as described previously, we present multiple measures to support our claim that memory drift makes an important contribution to behavior. Our two model-free metrics of drift reveal that entropy of reports decreased by an average of 52% (indicating more clustering) and mean absolute bias increased by an average of 21% from the short to the long delay. Furthermore, our three model-based metrics (AIC, BIC, and cross-validation) consistently reveal a significant impact of drift during the memory delay. Indeed, the cross-validation analyses (which we have improved by adding more fitting iterations to reduce the variance of this metric) reveals that, across all trials, the log-likelihood of the full model exceeds that of the model without memory drift by 11.8 in humans, 8.2 in monkey W, and 61.6 in monkey E. These values indicate that the full model is 1.3×10^5 , 3.6×10^3 , and 5.8×10^{26} times more likely than the model without memory drift, respectively (likelihood ratio).

Finally, we would like to note that a zero-delay condition is not necessary to separate encoding and memory drift. For our model-based analyses, we are able to dissociate encoding and memory effects by measuring error at multiple delay periods. By analogy, in simple linear regression, one does not need to

measure y when $x = 0$ in order to estimate an intercept. With regards to our model-free measures of drift (Figure S1), the absence of a zero-delay condition means that, if anything, we are underestimating the amount of drift introduced by memory. Furthermore, a strong interpretation of a 'zero-delay' condition in a continuous report paradigm can be problematic, as other forms of memory, such as iconic memory, may be involved.

Comment 2.2. line 72: "... suggesting that clustering is the result of a load-dependent dynamic process that unfolds over the course of the memory delay and not simply introduced at encoding" is ambiguous in this respect, possibly meaning that encoding did not introduce biases.

We thank the reviewer for pointing this out and have changed this line to "... that unfolds over the course of encoding and the memory delay".

Comment 3. There is still circularity in some of the main arguments, which has not been clarified in the revised manuscript. Figure 4 is presented as support for discrete dynamics, but the data reported in Figure 4 is mostly just validating the fit.

We apologize for the confusion, but Figure 4 shows the ability of the model to generalize to features of the data beyond those to which it is fit. We fit the model by maximizing a particular objective function (here, the likelihood of a response given the target color and delay length). Figure 4 shows that the resulting model can generalize to explain the clustering, bias, and precision of memory reports. These are separate features and there is no guarantee that any fit model will be able to recapitulate them (as described below), so there is no double-dipping or circular reasoning.

Thus, by showing the model can recapitulate so many features, Figure 4 provides evidence for discrete attractors in working memory.

Any possible fit of the data that includes the possibility of non-homogeneous reports will yield similar results. In particular, both the fact that reports cluster around fitted "attractors" (Fig. 4a) and that reports are less dispersed around these "attractors" (Fig. 4c) are a simple byproduct of having fitted correctly the nonhomogeneous set of reports with just any reasonable model. See the attached code `gaussianfit.py` or the model used by Bae et al. 2014, 2015. Authors responded to this criticism by mentioning an additional cross-validation approach, but this does not address my concern, it just shows that systematic biases are true biases and not due to random fluctuations, but not that the discrete dynamics model is particularly compelling.

As we detailed in our previous response, all other models that have currently been put forth in the literature, or by the reviewer, fail to capture all aspects of the behavior that we describe: the non-homogeneous reports, the observed bias in responses, the precision of responses, the time-dependent changes in these effects, the load-dependent changes in these effects, and the context-dependent changes in these effects. In our previous response and in our manuscript, we have ruled out several competitor models. First, we have shown that biased guessing (which allows for non-homogeneous responses) does not account for the observed changes in bias or precision (Figure S4). Second, other leading models of memory (e.g., variable precision), which are fit to the likelihood of responses just as is our model, are not able to reproduce the clustering of memory reports. Third, the model from Bae et al.

2015 cannot account for temporal evolution of memories and depends on linguistic encoding (not available in NHPs). Fourth, a mismatch between physical and perceptual space (the 'NHR' model from the reviewer) cannot capture bias or the correlation between mean and model report.

In contrast, our model reproduces these phenomenon (Figure 4, Figure 3, Figure 5) and provides a mechanistic, principled framework for why they occur. Future models may surpass the explanatory power of our model, in which case our understanding of working memory would be further improved.

The one piece of data in this figure that does support the discrete dynamics model is the bias (Fig. 4c,d). This is the main point and needs to be specifically stressed. As shown (and nicely interpreted) in their rebuttal, the NHR model predicts a different pattern of bias, and so does a modulated Gaussian model (gaussianfit.py). In my view, Fig. 4a,b,e,f belong more to Fig.2, where the model is presented and it is shown that it can fit the data (as many other models, previously shown by Bae et al. 2014, 2015). Instead, fig. 4 would be more about presenting the bias as the critical feature to look at to discriminate between alternative models, something it has not been shown before.

We agree with the reviewer that bias is diagnostic in distinguishing our model from some alternatives. However, other alternative models are excluded based on the other behavioral data (see above). So, we feel that presenting all data together will nicely summarize the explanatory power of our model and clearly define the features that should be matched by future (improved) models.

Comment 4.1 The new data with parametric variation of delay duration is very nice. However, the reporting of results should be improved. The model fit is now relegated to Fig. S6 but this is not in line with the main point of the manuscript, that seeks to establish the plausibility of a specific model. The model fit should be in the main figure, and its deviation from a straight line should be established statistically instead of resorting to a power law fit that lacks any specific motivation. Parameters for this model fit should also be reported. Also, there is an apparent discrepancy between the power law fit now reported in Fig. 5d and the (quite linear) model fit in Fig. S6. Is it correct to interpret that the data is consistent with a set of very shallow discrete attractors, approximating imperfectly a line attractor?

We thank the reviewer for pointing out this potential point of confusion. The apparent discrepancy between the power law fit and the model fit is due to the fact that the linearity test for discrete attractor dynamics is asymmetric. Discrete attractors may show linear or sub-linear increases in squared error, even when there are strong attractors (Kilpatrick et al, 2013; the ratio of drift to diffusion, not the absolute value of drift, determines sub-linearity). In contrast, a line attractor / diffusion process will *always* show a linear increase in squared error. Therefore, while a sub-linear increase in squared error is indicative of discrete attractors (and excludes a line attractor as an explanatory mechanism), a linear trend does not discriminate between discrete attractors and diffusion. Critically, the empirical data show evidence of a sub-linear increase, providing evidence against a diffusion process. Therefore, while there may be an apparent discrepancy between the fits, there is no discrepancy in interpretation.

That being said, we were curious as to what might cause the apparent discrepancy, given the strong evidence for sub-linear accumulation of error in the behavioral data. Through simulations, we found that a larger ratio of drift to diffusion allowed the model to capture this feature. This suggested that our

model fits were somehow overestimating the amount of diffusion or underestimating the drift during the memory delay. This could have arisen if the model was attempting to capture random error introduced at the response stage by lowering the drift-to-diffusion ratio during the memory delay (because of the error-correcting properties of attractors, decreasing drift increases random error). Therefore, we tested whether a model that included random error in the response was a better fit to the data. Indeed, this model was greatly preferred for our human subject data. However, the model was not preferred for the monkey data. This difference may reflect differences in response modality – humans scrolled through colors with a mouse, while monkeys saw the entire color wheel at once and made an eye movement. We have now updated all model-based human analyses in the manuscript using this better fitting model. We also detail this reasoning in the Methods (lines 554-566) .

Consistent with the idea that response error was being attributed to the memory delay, the updated model fits display a decrease in the strength of diffusion and an increase in the strength of drift during the memory delay. As predicted, this leads to a sublinear increase in error over time (Figure 7a). However, this sub-linearity is still not on the timescale of experiment 1b (Figure S6; thus not achieving our initial goal when we decided to include a response error). Therefore, to avoid confusion we now focus on how well the model fits the subjects' error, without discussing the sub-linearity (lines 157-161). We hope this is in line with the reviewer's general suggestion to focus on the fit of the model.

As suggested, we report parameter fits for this model (Figure S6c). These were comparable with those from experiment 1a.

Comment 5.1. The simulation results in Fig. 7b are surprising. It seems like the "drift+diffusion" model (the full model of fig. 3a) has very similar average mean error at the beginning of the memory period than the diffusion model (with neither encoding nor memory drift, i.e. similar to $\beta^*=0$ in fig. 3a when considering time $t=0$). How is that possible? Since the encoding period has the largest impact on drift (Table S2, Fig 6c,d), one would intuitively expect the "drift+diffusion" model to have a much larger mean error at the beginning of memory than the diffusion model. This can be clarified by showing in fig. 7a-b the simulation data from the encoding period, too (as in fig. 3a).

We thank the reviewer for noting this potential point of confusion. The error of the drift+diffusion and diffusion models are relatively similar at the start of the memory delay due to the fact that drift has both salutary and deleterious effects. For the drift+diffusion model, drift introduces bias but also counteracts diffusion during encoding. The diffusion model has no bias but experiences unchecked diffusion during encoding. These tradeoffs result in relatively similar error at the start of the memory delay.

As suggested by the reviewer, we have added a depiction of the encoding period to figure 7a (former figure 7b, magnified below, note error is not zero at time zero due to decoding error) and clarify this in the associated text (lines 197-199). We have removed former Figure 7a to make room for the new parametric analysis of target distributions described below, but we show the encoding period for this figure below for reference:

Comment 5.2. Also, there is no mention of how more frequent "attractor" colors have to be in order for the "colors biased" condition to make the drift+diffusion model better than the diffusion model at all times.

We thank the reviewer for this suggestion – we now study the effects of parametrically varying the proportion of attractor colors in Figure 7B. Performance is better for the drift + diffusion model for all t with at least 40% of targets drawn near attractor states.

Comment 6.1. The new analysis with short and long delay trials is very interesting. The comparisons between short/long and first/last third should be supported with an ANOVA-like analysis.

We thank the reviewer for this suggestion and now supplement our non-parametric difference tests with a non-parametric regression on lines 237-238. As expected, we observe a main effect of delay length ($p = 0.026$), no effect of epoch (first or last third of trials, $p = 0.45$), and a delay \times epoch interaction ($p = 0.039$).

Comment 6.2. Surprisingly, for short delay trials there appears not to be much change between the last third and the first third of the session. This suggests that the "context adaptation" mechanism affects only the memory drift component, and not the encoding drift component of the model.

We agree with the reviewer that this difference is interesting and suggests adaptation to new contexts may involve multiple neural mechanisms with different timescales. We are very interested in following up on this in the future, but feel that it is outside of the domain of the current manuscript.

Based on this result, memory and encoding drifts will typically respond to different sets of attractors (encoding drift will be fixed, while memory drift will adapt). How does this influence the reasoning in Fig. 7b? Intuitively, the "colors biased" condition will be adapted to the memory drift, but not to the encoding drift, so there will always be a behavioral cost to using the "drift+diffusion" model for short memory delays.

Our results from Experiment 1 suggest that the encoding and memory dynamics will eventually align with one another. This is reflected in the fact that attractor locations were the same for short and long duration trials in these experiments (Figure S2). So, while abrupt changes in the statistics of the environment may initially affect the drift dynamics underlying memory it seems that drift during encoding will eventually catch up. This supports the idea that there may be multiple neural mechanisms at play. We thank the reviewer for noting this and now discuss this possibility in the discussion (line 294-296).

Comment 7. The last paragraph of results concludes that clustering reduction is consistent with weakening of baseline attractors, but it could be solely the result of strengthening of uncorrelated context-defined attractors (as shown in Fig. 7). As used in Fig. 7, the real test for strengthening or weakening of attractors is the response bias at the location of the attractor. It is not justified not to report this direct test of the modulation of baseline attractor strength. The prediction to be tested is that bias slopes around baseline attractors should get less negative in the last third compared to the first third of the session.

We agree with the reviewer and thank them for pointing this out. We previously reported that responses continue to cluster at baseline locations (Figure S8), suggesting that attractors are updated with a slow learning rate, ideal for extracting statistical regularities. As suggested by the reviewer, we tested whether the slope of the bias around baseline attractors significantly changed during the task. It did not (change = -0.018, $p=0.570$, bootstrap). This further supports the idea that baseline attractors are resilient, at least for the timescale of a couple hundred trials. We have added this new analysis to the text (line 256-257).

Minor:

- model parameters sometimes have an n subindex, sometimes an L subindex.

Thank you for noticing this, we now use consistent notation

- Eq. (12): alpha should vanish for load 1 trials. The current notation does not reflect that.

We now state this explicitly on line 536

- Fig. R6 is incorrect. The NHR model does not show a linear increase, as readily computed with simulations. Also, the discrete attractor model curve is far from the data fit (Fig. S6). In practice, these two models are indistinguishable in this graph.

We have removed the sub-linearity analysis from the manuscript and so this is no longer relevant to the current manuscript.

- authors resolve one problem in interpretation of Fig. 7 by resorting to debriefing data from participants. The specific question that participants responded to should be literally stated in Methods.

We thank you for the suggestion, we now include this on lines 407-410.

Reviewers' Comments:

Reviewer #4:

Remarks to the Author:

Despite some discrepancies that I still have with the authors in relationship to their interpretations of some graphs, the manuscript is now much improved. It is also very commendable that the data is made available so readers will be able to evaluate the results quantitatively themselves. I only have two more requests that should be easy to implement and would make more clear some of the points of the manuscript that I was struggling with:

1) The authors convincingly argue that there are multiple other models that would give different results than what is now shown in Figure 4. This is indeed a reasonable argument to maintain this figure as a result. However, this key argument is not in the manuscript now, as 2 models are presented (Fig. S4, S5) but not the "leading models of memory (e.g., variable precision), which are fit to the likelihood of responses just as is our model, are not able to reproduce the clustering of memory reports". I suggest that the text includes mention to these models, too. By presenting these leading fitted models and their predictions, the point of Fig. 4 will now be clear to this reviewer and the reader of the manuscript.

2) I think that removing the focus on the sublinearity of the relationship between error and delay is reasonable. Especially considering that a diffusion process consistent with a line attractor also presents a sublinear law when limited to a circular topology (see Fig. 7a). A sublinear trend does not by itself discard a line attractor on a ring topology, contrary to what the rebuttal now states. What is relevant is how sublinear it is. It remains a quantitative issue which the empirical data (Fig. S6) does not seem to clearly disambiguate, as the sublinearity observed in the data is weak. In this sense, an analysis with this data in the line of what is done in Fig. 3 for Experiment 1a would be informative, and would provide the model comparative for this dataset that is applied to the rest of the data in this manuscript.

Reviewer 4 comments:

Despite some discrepancies that I still have with the authors in relationship to their interpretations of some graphs, the manuscript is now much improved. It is also very commendable that the data is made available so readers will be able to evaluate the results quantitatively themselves. I only have two more requests that should be easy to implement and would make more clear some of the points of the manuscript that I was struggling with:

1) The authors convincingly argue that there are multiple other models that would give different results than what is now shown in Figure 4. This is indeed a reasonable argument to maintain this figure as a result. However, this key argument is not in the manuscript now, as 2 models are presented (Fig. S4, S5) but not the "leading models of memory (e.g., variable precision), which are fit to the likelihood of responses just as is our model, are not able to reproduce the clustering of memory reports". I suggest that the text includes mention to these models, too. By presenting these leading fitted models and their predictions, the point of Fig. 4 will now be clear to this reviewer and the reader of the manuscript.

We thank the reviewer for pointing out this oversight. We now mention that variable precision models and slot-based working memory models do not specifically predict variation in clustering, bias, or precision across color space (lines 274-276).

2) I think that removing the focus on the sublinearity of the relationship between error and delay is reasonable. Especially considering that a diffusion process consistent with a line attractor also presents a sublinear law when limited to a circular topology (see Fig. 7a). A sublinear trend does not by itself discard a line attractor on a ring topology, contrary to what the rebuttal now states. What is relevant is how sublinear it is. It remains a quantitative issue which the empirical data (Fig. S6) does not seem to clearly disambiguate, as the sublinearity observed in the data is weak. In this sense, an analysis with this data in the line of what is done in Fig. 3 for Experiment 1a would be informative, and would provide the model comparative for this dataset that is applied to the rest of the data in this manuscript.

As suggested by the reviewer, we now include a model comparison for the Experiment 1b data and now provide this on lines 408-410.

We should note that the range of errors discussed in the manuscript are all well outside the range where wrapping due to circular topology would be an issue. Therefore, in this domain, a pure diffusion process does predict a linear increase in error with time.